# Bridging Knowledge Gaps in Small Cell Lung Cancer: Data, Challenges and Priorities

**DOI:** 10.3390/curroncol32100536

**Published:** 2025-09-25

**Authors:** Chiara Catania, Priscilla Cascetta, Alessandro Russo, Emily Governini, Marzia Bendoni, Alice Laffi, Ilaria Piloni, Fabio Conforti, Laura Pala, Emilia Cocorocchio, Giovanni Ceresoli, Marzia Locatelli, Daniele Laszlo, Flaminia Facella, Tommaso De Pas

**Affiliations:** 1Division of Oncology, Humanitas Gavazzeni, 24125 Bergamo, Italyalice.laffi@gavazzeni.it (A.L.); tommaso.depas@gavazzeni.it (T.D.P.); 2Department of Biomedical Sciences, Humanitas University, Pieve Emanuele, 20072 Milan, Italy; 3Medical Oncology Department, Humanitas Istituto Catanese, Misterbianco, 95045 Catania, Italy

**Keywords:** small cell lung cancer, immunotherapy, molecular subtype, precision oncology

## Abstract

Small Cell Lung Cancer (SCLC) is a highly aggressive neuroendocrine tumor comprising 15% of lung cancers. Despite initial sensitivity to chemotherapy, immunotherapy and radiotherapy, relapse is nearly universal due to acquired resistance. Molecular profiling has revealed four transcription factor–driven subtypes (ASCL1, NEUROD1, POU2F3, YAP1), yet clinical application remains limited. Immunotherapy offers modest survival gains, with only 10–15% of patients surviving beyond five years. Emerging agents like tarlatamab and ifinatamab deruxtecan show promise, but predictive biomarkers are urgently needed. Future research must focus on precision medicine through integrated biomarker analysis and immune profiling to improve patient outcomes.

## 1. Introduction

Small Cell Lung Cancer (SCLC) is an aggressive neuroendocrine malignancy accounting for approximately 15% of all lung cancers. Only 30% of cases present as limited-stage small cell lung cancer (LS-SCLC), while 70% are diagnosed as extensive-stage (ES-SCLC) [1]. It is characterized by rapid progression, early metastatization, and a high prevalence of circulating tumor cells, contributing to its poor prognosis. Despite initial responsiveness to chemotherapy, immunotherapy and radiotherapy, nearly all patients experience relapse due to acquired resistance. Recent molecular studies have identified distinct subtypes of SCLC based on transcription factor expression, including ASCL1, NEUROD1, POU2F3, and YAP1-driven subtypes [2]. These subtypes exhibit unique biological behaviors and therapeutic vulnerabilities, but their clinical implementation still remain a major challenge and, therefore, clinical classification remains largely homogeneous. Immunotherapy has shown modest survival benefits, with only a subset of patients responding to immune checkpoint inhibitors and a 5-years survival rate still estimated to be around 10–15% [3].

## 2. Classification of Pulmonary Neuroendocrine Tumors According to Grading (G)

Pulmonary neuroendocrine tumors are classified based on histological grading (G), which reflects their degree of differentiation, mitotic activity, and presence of necrosis. The following classification is widely accepted in clinical and pathological practice:

Typical carcinoid (TC): Low-grade (G1), well-differentiated neuroendocrine tumor characterized by the absence of necrosis and fewer than 2 mitoses per 2 mm^2^.

Atypical carcinoid (AC): Intermediate grade (G2), well-differentiated neuroendocrine tumor with focal necrosis and between 2 and 10 mitoses per 2 mm^2^.

Large Cell Neuroendocrine Carcinoma (LCNEC): High-grade, poorly differentiated neuroendocrine carcinoma with extensive necrosis and more than 10 mitoses per 2 mm^2^.

Small Cell Lung Cancer (SCLC): High-grade, poorly differentiated neuroendocrine carcinoma characterized by aggressive clinical behavior and high mitotic index [4,5].

Extensive Small Cell Lung Cancer (SCLC): Approximately 70% of small cell lung cancers (SCLC) present with extensive-stage disease, while 30% are diagnosed at a limited stage, and among these, only 2–3% are detected at stage I–II. In advanced-stage SCLC, current therapeutic options include chemotherapy, immunotherapy, and radiotherapy, along with novel agents that have emerged in recent years. The integration of these treatments has led to significant improvements in efficacy and survival outcomes in this highly aggressive malignancy, where until a few years ago, chemotherapy and radiotherapy were the only available options. Recent transcriptomic analyses of clinical trial samples have further refined our understanding of SCLC heterogeneity. In a pivotal study published by Barzin Y. Nabet et al. in Cancer Cell, four molecular subsets of SCLC were identified, offering new insights into tumor biology and therapeutic responsiveness. Notably, immune-inflamed subsets were found to exhibit both neuroendocrine (NE) and non-neuroendocrine (non-NE) phenotypes, underscoring the plasticity of SCLC. These subsets also demonstrated heterogeneous immune cell composition, which was associated with variable clinical outcomes. Importantly, NE tumors characterized by high T-cell infiltration and low macrophage density showed enhanced benefit from anti-PD-L1 immunotherapy combined with chemotherapy, suggesting a potential biomarker-driven approach to treatment selection [6,7].

### 2.1. Rationale for the Addition of Immunotherapy to Chemotherapy

The integration of immunotherapy with standard chemotherapy in the treatment of extensive-stage small cell lung cancer (SCLC) is supported by a multifaceted rationale aimed at enhancing anti-tumor immune responses. The key mechanisms include:Increasing T-cell infiltration into the tumor microenvironment, thereby improving immune surveillance and the potential for tumor cell eradication.Enhancing the functional capacity of effector T cells, which are critical for mediating cytotoxic responses against tumor cells.Improving the recognition of tumor-associated antigens by T cells, facilitating a more targeted and effective immune response.Eliminating immunosuppressive cell populations, such as regulatory T cells (Tregs), myeloid-derived suppressor cells (MDSCs), and M2-polarized macrophages, which collectively contribute to immune evasion by the tumor.Inducing immunogenic cell death, a process by which dying tumor cells release danger signals and antigens that stimulate a robust immune response.Promoting the maturation and activation of dendritic cells, thereby enhancing antigen presentation and the priming of naïve T cells.Targeting and depleting regulatory T cells, which play a central role in maintaining an immunosuppressive tumor microenvironment [8].

These immunomodulatory effects provide a strong biological rationale for combining immune checkpoint inhibitors, such as atezolizumab, with cytotoxic chemotherapy in the first-line treatment of extensive-stage SCLC.

### 2.2. IMpower133: First-Line Atezolizumab Plus Chemotherapy in Extensive-Stage Small Cell Lung Cancer (ES-SCLC)

The IMpower133 phase III trial evaluated the efficacy of adding atezolizumab, a PD-L1 inhibitor, to standard chemotherapy with carboplatin and etoposide (CP/ET) in treatment-naive patients with extensive-stage small cell lung cancer (ES-SCLC). Patients received four induction cycles of CP/ET combined with either atezolizumab or placebo. Those who achieved disease control continued with maintenance therapy—atezolizumab in the experimental arm or placebo in the control arm—until disease progression or unacceptable toxicity. Eligible participants had histologically or cytologically confirmed ES-SCLC and underwent brain imaging at baseline. Patients with treated, asymptomatic brain metastases were included, while those with active or untreated central nervous system involvement were excluded. Stratification at enrollment was based on sex, ECOG performance status (0 vs. 1), and presence of brain metastases. Tumor samples were collected using various techniques for exploratory biomarker analyses, including PD-L1 expression and blood-based tumor mutational burden (bTMB), in the biomarker-evaluable population. The co-primary endpoints were overall survival (OS) and progression-free survival (PFS). At the time of the primary analysis, with a median OS follow-up of 13.9 months, the atezolizumab arm showed a significant survival benefit. Median OS was 12.3 months with atezolizumab plus CP/ET, compared to 10.3 months with placebo plus CP/ET (hazard ratio [HR] = 0.70; 95% CI, 0.54–0.91; *p* = 0.007). Median PFS was also improved (5.2 vs. 4.3 months; HR = 0.77; 95% CI, 0.62–0.96; *p* = 0.02) [8].

These results led to the approval of atezolizumab in combination with CP/ET as a first-line treatment for ES-SCLC by both the U.S. Food and Drug Administration (FDA) and the European Medicines Agency (EMA). A preplanned updated OS analysis, published in the Journal of Clinical Oncology (2021), was conducted after an additional 9 months of follow-up, extending the median OS follow-up to 22.9 months [8]. This analysis provided further evidence of the long-term benefit and safety of the regimen. It also included exploratory evaluations of OS by bTMB levels and PD-L1 expression, as well as patterns of disease progression, offering valuable insights into potential predictive biomarkers and mechanisms of resistance. At the 12-month mark, overall survival (OS) was 51.9% in the atezolizumab plus CP/ET group, reflecting a 12.9% improvement over the 39.0% observed in the placebo group. This survival advantage persisted at 18 months, with 34.0% of patients in the atezolizumab arm still alive, compared to 21.0% in the control arm—an absolute difference of 13.0%. These findings align with the primary analysis and confirm that the addition of atezolizumab consistently improved OS across most patient subgroups. In the updated analysis of the intent-to-treat (ITT) population, the median progression-free survival (PFS) was 5.2 months (95% CI, 4.4–5.6) in the group receiving atezolizumab combined with carboplatin and etoposide, compared to 4.3 months (95% CI, 4.2–4.5) in the chemotherapy plus placebo group. This corresponds to a hazard ratio (HR) of 0.77 (95% CI, 0.63–0.95). In the updated analysis of the IMpower133 trial patterns of progression were also analyzed in detail. In the atezolizumab and placebo arms, respectively, progression occurred at pre-existing lesions in 57.7% and 64.9% of patients, at new lesions in 42.8% and 49.0%, and at both new and existing sites in 20.9% and 28.2%. The distribution of progression across specific organs was largely comparable between treatment groups. The most frequent sites of new lesions—each occurring in at least 10% of patients—included the central nervous system (CNS), lungs, lymph nodes, and liver, with similar incidence rates across both arms.

Among patients who received prophylactic cranial irradiation (PCI)—22 in each treatment group—progression at existing lesions was reported in 68.2% of those in the atezolizumab arm and 77.3% in the placebo arm. New lesions developed in 45.5% and 54.5% of patients, respectively, while progression at both new and existing sites occurred in 27.3% and 31.8%. Notably, only one patient in each group developed new brain metastases following PCI. In contrast, among patients who did not receive PCI, new brain lesions emerged in 21.1% of those treated with atezolizumab and 22.2% of those in the placebo group. These findings suggest that while the addition of atezolizumab modestly delays disease progression, the overall patterns and sites of progression remain largely consistent between treatment arms, including in the context of CNS involvement and PCI administration. Although PD-L1 expression is a well-established biomarker in non-small cell lung cancer, its role in small cell lung cancer (SCLC) remains unclear. In clinical trials evaluating immune checkpoint inhibitors such as atezolizumab, PD-L1 expression—particularly on tumor cells—was generally low and inconsistently associated with treatment outcomes. While some patients with PD-L1 expression < 1% appeared to benefit from immunotherapy, no consistent survival advantage was observed in those with higher PD-L1 levels. Moreover, PD-L1 was predominantly expressed on immune cells rather than tumor cells, further complicating its predictive value. These findings suggest that PD-L1 alone is not a reliable biomarker for guiding immunotherapy in SCLC, highlighting the need for alternative markers such as tumor mutational burden or molecular subtyping. This analysis confirmed that patients experienced clinical benefit from treatment regardless of whether their blood tumor mutational burden (bTMB) was classified as high or low, based on the predefined thresholds of 10 and 16. These findings suggest that bTMB alone may not be sufficient to predict which patients will respond best to therapy. Notably, there was only partial overlap between patients with high bTMB and those with elevated PD-L1 expression. Specifically, fewer than 56% of PD-L1–positive patients (defined as ≥1% expression on tumor or immune cells) also had high bTMB (≥10). This indicates that bTMB and PD-L1 capture distinct biological subgroups, and neither biomarker can fully substitute for the other in predicting response to immunotherapy. Overall, the data suggest that clinical benefit was observed in patients with ES-SCLC treated with 1L atezolizumab plus CP/ET, independent of bTMB or PD-L1 biomarker status. Five-Year Survival Outcomes from the IMpower133 Trial: Atezolizumab Plus Chemotherapy in ES-SCLC. A recent publication by Martin Reck and colleagues in Lung Cancer reports the five-year survival outcomes of patients with extensive-stage small-cell lung cancer (ES-SCLC) enrolled in the IMpower133 trial. In this phase III, randomized, double-blind study, patients received four cycles of carboplatin and etoposide in combination with the anti–PD-L1 monoclonal antibody atezolizumab, followed by maintenance therapy with atezolizumab or placebo [9]. A subsequent rollover study, IMbrella A, enrolled 18 patients previously treated with atezolizumab in IMpower133 between December 2019 and July 2020 [3]. At the five-year mark, the median overall survival (OS) for patients in the atezolizumab maintenance arm was 12.3 months, compared to 10.3 months in the placebo group (3). Notably, 12% of patients in the experimental arm were still alive at five years. These results represent a substantial improvement over historical five-year survival rates of approximately 2% for patients with ES-SCLC treated with chemotherapy alone. These findings suggest the existence of a subset of patients who derive long-term benefit from immunotherapy. Future research should aim to identify predictive biomarkers to better select patients most likely to respond durably to immune checkpoint inhibition.

### 2.3. Durvalumab in Extensive-Stage Small-Cell Lung Cancer: Long-Term Outcomes from the CASPIAN Trial

The CASPIAN study is a phase III, open-label, randomized clinical trial evaluating the efficacy of adding immunotherapy to standard chemotherapy in patients with extensive-stage small-cell lung cancer (ES-SCLC). In the experimental arm, patients received four cycles of platinum-based chemotherapy (cisplatin or carboplatin) combined with etoposide and durvalumab. Those without disease progression continued with durvalumab maintenance until unacceptable toxicity or disease progression. The control arm received up to six cycles of chemotherapy alone, with optional prophylactic cranial irradiation (PCI) following treatment. A third arm included a combination of tremelimumab (an anti–CTLA-4 antibody), durvalumab, and chemotherapy for four cycles, followed by durvalumab maintenance [3]. An updated analysis at 24 months confirmed a mOS of 12.9 months with durvalumab versus 10.4 months in the control arm (HR: 0.75; 95% CI: 0.62–0.91). The most recent update, published in *ESMO Open* (2022), reported 3-year follow-up data: mOS remained at 12.9 months in the durvalumab arm versus 10.5 months in the control group (HR: 0.71; 95% CI: 0.60–0.86). At three years, 17.6% of patients treated with chemo-immunotherapy were still alive, compared to only 5.8% in the chemotherapy-only group [10,11].

These findings support the long-term survival benefit of incorporating durvalumab into first-line treatment for ES-SCLC, establishing it as a new standard of care.

### 2.4. ADRIATIC Trial: Redefining the Standard of Care in Limited-Stage SCLC with Durvalumab Consolidation

The ADRIATIC trial (NCT03703297) is a pivotal, global, phase III, randomized, double-blind, placebo-controlled study designed to evaluate the efficacy and safety of durvalumab, with or without tremelimumab, as consolidation therapy in patients with limited-stage small-cell lung cancer (LS-SCLC) who had not experienced disease progression following concurrent chemoradiotherapy (cCRT). This trial addresses a significant unmet clinical need in limited-stage small-cell lung cancer (LS-SCLC), a disease setting in which therapeutic progress has historically been constrained. Eligible patients included individuals with unresectable stage I to III LS-SCLC and a World Health Organization (WHO) performance status of 0 or 1. All participants received a standardized induction regimen comprising four cycles of platinum-based chemotherapy—either carboplatin or cisplatin—in combination with etoposide and concurrent thoracic radiotherapy. Radiotherapy was administered either as 60–66 Gy over six weeks or as 45 Gy delivered twice daily over three weeks, in accordance with institutional standards. Prophylactic cranial irradiation (PCI) was permitted prior to randomization, reflecting real-world clinical practice. Following completion of chemoradiotherapy, patients who remained progression-free were randomized in a 1:1:1 ratio to receive maintenance therapy for up to 24 months or until disease progression or unacceptable toxicity. The three treatment arms included durvalumab monotherapy (durvalumab 1500 mg every four weeks plus placebo), combination immunotherapy (durvalumab 1500 mg plus tremelimumab 75 mg every four weeks), and placebo control (placebo plus placebo) [12]. This design allowed for a rigorous evaluation of both monotherapy and combination immunotherapy strategies in a well-defined LS-SCLC population.

### 2.5. Key Efficacy Outcomes

The ADRIATIC trial represents a landmark phase III study that has, for the first time, demonstrated a statistically significant and clinically meaningful survival benefit with immunotherapy in patients with limited-stage small-cell lung cancer (LS-SCLC) who did not progress following concurrent chemoradiotherapy (cCRT). At the time of the first planned interim analysis, with a data cutoff of 15 January 2024, a total of 730 patients had been randomized to receive either durvalumab or placebo. The median follow-up for overall survival (OS) was 37.2 months. The results showed a substantial improvement in OS for patients treated with durvalumab. Specifically, the median OS reached 55.9 months in the durvalumab arm, compared to 33.4 months in the placebo group. This corresponds to a hazard ratio (HR) of 0.73, indicating a 27% reduction in the risk of death, with the result achieving statistical significance (*p* = 0.0104). Notably, the survival benefit was sustained over time, with 24- and 36-month OS rates of 68.0% and 56.5% in the durvalumab group, respectively, compared to 58.5% and 47.6% in the placebo arm. Progression-free survival (PFS) was also significantly prolonged with durvalumab. The median PFS was 16.6 months versus 9.2 months in the placebo group (HR: 0.76; *p* = 0.0161), with higher PFS rates observed at both 18 and 24 months. Importantly, the survival benefit was consistent across all predefined subgroups, including age, sex, disease stage, chemotherapy regimen, and use of prophylactic cranial irradiation (PCI), underscoring the robustness of the findings. From a safety perspective, durvalumab was generally well tolerated. The incidence of grade 3–4 adverse events was comparable between the two arms (24.3% vs. 24.2%), although treatment discontinuation due to adverse events was slightly higher in the durvalumab group (16.3% vs. 10.6%). Pneumonitis, including radiation pneumonitis, occurred more frequently with durvalumab (38.0% vs. 30.2%), but the rate of severe (grade 3–4) events remained low and similar between groups [12]. In conclusion, the ADRIATIC trial establishes durvalumab as a new standard of care in LS-SCLC following cCRT. The magnitude of the overall survival benefit—an absolute gain of over 22 months—is unprecedented in this setting and marks a significant advancement in the treatment of a historically challenging disease (Figure 1).

In contrast, the use of immunotherapy concurrent with chemo-radiotherapy instead as consolidation therapy did not improve the OS in LS-SCLC in the phase 3 NRG Oncology/Alliance LU005 [13]. similarly to that observed in NSCLC with the PACIFIC-2 study [14], raising the question that concomitant immunotherapy with chemo-radiation might be associated with lower efficacy due to interference on immunotherapy activity.

### 2.6. Triple Modality in SCLC: Is There a Rationale for Combining Chemotherapy, Radiotherapy, and Immunotherapy?

#### 2.6.1. Biological and Clinical Rationale Behind KEYLYNK LD-SCLC

The KEYLYNK LD-SCLC trial explores a novel therapeutic strategy by integrating chemotherapy, immunotherapy, and radiotherapy in patients with limited-stage small-cell lung cancer (LS-SCLC) [15]. The rationale for this approach stems from emerging evidence that combining immune checkpoint inhibitors with DNA damage-inducing agents may enhance antitumor efficacy.

Pembrolizumab, an anti–PD-1 monoclonal antibody, has shown promising activity when administered concurrently with chemoradiotherapy in other thoracic malignancies, such as non-small-cell lung cancer (NSCLC), where it demonstrated high objective response rates and durable outcomes.

Olaparib, a PARP inhibitor, is included in the maintenance phase based on its potential to synergize with immunotherapy. Preclinical studies suggest that PARP inhibition can upregulate PD-L1 expression and activate STING-dependent immune pathways, thereby sensitizing tumors to checkpoint blockade—even in the absence of BRCA mutations.

This combination aims to exploit the immunogenic effects of radiotherapy and chemotherapy, which may increase neoantigen release and immune cell infiltration, while olaparib may further enhance immune recognition and persistence of response. The trial design reflects a growing interest in triplet therapy to overcome the immunosuppressive tumor microenvironment typical of SCLC.

If successful, KEYLYNK LD-SCLC could redefine the standard of care for LS-SCLC by identifying a subset of patients who benefit from long-term disease control through multimodal treatment.

#### 2.6.2. KEYLYNK LD-SCLC: Is Triple Modality Therapy the Key?

The KEYLYNK LD-SCLC trial is enrolling patients with limited-stage small-cell lung cancer (LS-SCLC), specifically those classified as stage I to III according to AJCC 8th edition, with an ECOG performance status of 1, and who are treatment-naïve. In the experimental arm, patients receive etoposide and platinum-based chemotherapy (CT) combined with pembrolizumab (an anti–PD-1 immune checkpoint inhibitor) starting from cycle 1. From cycle 2 to cycle 4, standard thoracic radiotherapy (RT) is added to the chemo-immunotherapy regimen. Following cycle 4, patients undergo prophylactic cranial irradiation (PCI), and from cycle 5 onward, they continue with maintenance therapy consisting of pembrolizumab plus olaparib (a PARP inhibitor) or pembrolizumab plus placebo. The control arm follows a similar structure: patients receive chemotherapy and thoracic radiotherapy, followed by PCI after cycle 4, and then continue with placebo plus placebo as maintenance [15]. This ongoing study aims to evaluate whether the integration of chemotherapy, immunotherapy, and radiotherapy can improve outcomes in LS-SCLC, a setting where durable responses remain a significant clinical challenge. Whether trimodality treatment truly represents the key to improved outcomes in ES-SCLC remains to be confirmed by ongoing and future studies. While preliminary results are promising, this approach is undoubtedly intensive in terms of toxicity and may not be feasible for all patients. Careful selection will therefore be essential to identify those who are most likely to benefit. More treatment may indeed lead to better outcomes, but therapeutic strategies must always balance efficacy with tolerability, ensuring that regimens can be safely administered—at least to a well-defined subset of this fragile population.

### 2.7. Early Maintenance Therapy in ES-SCLC: Insights from the IMforte Trial

The IMforte trial (NCT05091567) is a global, phase III, randomized, open-label study evaluating the efficacy of lurbinectedin plus atezolizumab versus atezolizumab alone as first-line maintenance therapy in patients with extensive-stage small-cell lung cancer (ES-SCLC) who did not progress after induction chemotherapy with carboplatin, etoposide, and atezolizumab [16]. A total of 660 treatment-naïve patients received four 21-day cycles of induction therapy. Of these, 483 patients without disease progression were randomized in a 1:1 ratio to receive maintenance treatment with either the combination of lurbinectedin and atezolizumab or atezolizumab monotherapy [2]. The trial demonstrated a statistically significant improvement in both progression-free survival (PFS) and overall survival (OS) in the combination arm. Median PFS was 5.4 months with the combination versus 2.1 months with atezolizumab alone (HR 0.54; 95% CI: 0.43–0.67; *p* < 0.0001), while median OS was 13.2 months versus 10.6 months, respectively (HR 0.73; 95% CI: 0.57–0.95; *p* = 0.0174) [2]. Importantly, these findings support the anticipation of second-line agents such as lurbinectedin into the first-line maintenance setting, potentially redefining the treatment paradigm for ES-SCLC. While the combination therapy was associated with a higher incidence of grade 3/4 treatment-related adverse events (25.6% vs. 5.8%), no new safety signals were observed. The IMforte trial is the first global phase III study to demonstrate both OS and PFS benefits with a first-line maintenance strategy in ES-SCLC, highlighting the clinical value of early integration of second-line agents to delay disease progression and improve survival outcomes.

### 2.8. Shorter or Longer? Defining the Optimal Chemotherapy Course with Immunotherapy in SCLC

The optimal number of chemotherapy cycles when combined with immune checkpoint inhibitors (ICIs) in extensive-stage small-cell lung cancer (ES-SCLC) remains a critical clinical question. While pivotal trials such as IMpower133 and CASPIAN established four cycles of platinum-etoposide chemotherapy followed by maintenance immunotherapy as the standard, emerging data from the MAURIS and LUMINANCE studies suggest that extending chemotherapy to five or six cycles may offer additional benefit in selected patients.

The MAURIS study, a multicenter phase III trial conducted in Italy, demonstrated that patients receiving five to six cycles of induction chemotherapy in combination with atezolizumab had significantly improved outcomes compared to those receiving four or fewer cycles [17]. Specifically, patients who received ≥6 cycles had a median overall survival (OS) of 18.5 months versus 13.1 months in those receiving fewer cycles (*p* < 0.001), and a median progression-free survival (PFS) of 8.0 months versus 5.0 months (*p* = 0.002). Subgroup analyses showed that the survival benefit was particularly evident in patients ≤ 65 years, males, and those without liver or bone metastases [1]. Similarly, the LUMINANCE study, a phase 3b trial evaluating durvalumab plus platinum-etoposide, reported that patients receiving five or more cycles of chemotherapy had superior outcomes compared to those receiving only four [18]. The overall response rate (ORR) was 81.2% in the ≥5-cycle group versus 47.0% in the 1–4 cycle group. Median PFS was 6.5 months versus 4.6 months, and median OS was not reached in the ≥5-cycle group compared to 10.7 months in the 1–4 cycle group [2]. Importantly, the safety profile remained manageable, with a lower incidence of grade ≥ 3 adverse events in patients receiving extended chemotherapy, likely reflecting a selection of patients with better tolerance.

These findings suggest that extending chemotherapy beyond four cycles in combination with immunotherapy may be beneficial for a subset of patients with ES-SCLC, particularly those with good performance status and limited comorbidities. However, the potential for cumulative toxicity and the need for individualized treatment decisions remain critical considerations.

### 2.9. Challenges, Unmet Needs, and the Path Toward Precision

Despite the transformative impact of immunotherapy in small cell lung cancer (SCLC), which has led to meaningful improvements in overall survival and progression-free survival, the field continues to face substantial challenges. One of the most pressing is the lack of clarity around which patients benefit most from these therapies. For instance, what defines the approximately 12% of patients who achieve long-term survival beyond five years? Understanding the biological and clinical features of these exceptional responders is critical to advancing precision medicine in SCLC.

To address this, future research must prioritize large, well-designed studies that integrate tumor tissue analysis, blood-based biomarkers, and immune profiling. Comprehensive and standardized evaluation of PD-L1 expression, tumor mutational burden (TMB), DLL3, CD3, and other emerging markers will be essential in uncovering predictive signatures and guide treatment decisions. Only through such multi-dimensional approaches can we begin to unravel the complex biology of SCLC and improve patient selection for immunotherapy. This need is further underscored by the rapid development of novel therapeutic agents. Tarlatamab, a DLL3-targeting bispecific T-cell engager, has recently received FDA approval for second-line treatment and is under investigation in the first-line setting combined with chemotherapy [19]. Likewise, B7-H3–targeting agents, such as ifinatamab deruxtecan, are showing promise in early-phase trials. In a Phase I/II study in patients with refractory SCLC, Authors reported an ORR: 52.4% with mPFS: 5.6 months and mOS12.2 months. Safety profile was manageable, with grade ≥3 adverse events in 36.4% of patients. Of interest, no significant correlation between B7-H3 expression and clinical efficacy was found [20]. A global, multicenter, randomized, open-label phase 3 trial (IDeate-Lung02) is currently underway to evaluate the efficacy and safety of the investigational antibody-drug conjugate (ADC) ifinatamab deruxtecan (I-DXd) versus standard second-line chemotherapy in patients with relapsed small-cell lung cancer (SCLC) following progression on first-line platinum-based therapy. The study randomizes eligible patients to receive either: Ifinatamab deruxtecan (I-DXd) at 12 mg/kg intravenously every three weeks, or Physician’s choice of chemotherapy, including topotecan, amrubicin, or lurbinectedin. This trial represents a significant step toward expanding treatment options in the second-line setting, where current therapies offer limited benefits. If successful, I-DXd could become the first B7-H3–directed therapy approved for SCLC. The toxicity profiles of tarlatamab and ifinatamab deruxtecan (I-DXd) differ significantly, reflecting their distinct mechanisms of action and therapeutic classes. Tarlatamab is associated with acute immune-mediated toxicities requiring specialized monitoring protocols, particularly for CRS and ICANS. I-DXd presents a more traditional cytotoxic profile, with manageable gastrointestinal and hematologic side effects, but carries a risk of ILD, necessitating pulmonary vigilance. These differences underscore the importance of patient selection and toxicity management strategies tailored to each agent’s pharmacologic profile. Tarlatamab (Bispecific T-cell Engager (DLL3-targeted)) Immune-related toxicities are predominant: Cytokine Release Syndrome (CRS): Reported in up to 72.7% of patients in real-world settings, compared to 51% in clinical trials [1]. Immune Effector Cell–Associated Neurotoxicity Syndrome (ICANS): Occurred in 40.9% of patients in real-world cohorts, with higher grades observed in those with untreated brain metastases [19]. In May 2024, the FDA granted approval for tarlatamab in this setting, marking a significant advancement in the therapeutic landscape of SCLC. The DeLLphi-304 phase 3 trial evaluated tarlatamab—a bispecific T-cell engager targeting DLL3—versus standard second-line chemotherapy (topotecan, lurbinectedin, or amrubicin) in patients with extensive-stage small-cell lung cancer (ES-SCLC) who had progressed following platinum-based chemotherapy and immunotherapy [21]. The study demonstrated a statistically significant improvement in overall survival (median OS: 13.6 vs. 8.3 months; HR: 0.60; *p* < 0.001) and progression-free survival (median PFS: 4.2 vs. 3.7 months; HR: 0.71; *p* = 0.002), alongside better patient-reported outcomes and a more favorable safety profile. Notably, grade ≥3 treatment-related adverse events occurred in 27% of patients receiving tarlatamab compared to 62% in the chemotherapy arm. These findings support the integration of tarlatamab as a new standard of care in the second-line setting for ES-SCLC. This consensus was further supported by the Delphi panel on SCLC management, which included international experts in thoracic oncology [22]. Recent advances in the molecular characterization of small-cell lung cancer (SCLC) have led to the identification of distinct transcriptional subtypes, defined by the differential expression of surface antigens such as SEZ6, DLL3, B7-H3 (CD276), TROP2 (TACSTD2), and CEACAM5. These markers are being actively explored as therapeutic targets to enable subtype-specific treatment strategies using antibody-drug conjugates (ADCs), bispecific T-cell engagers (BiTEs), and other targeted agents (Figure 2). Among these, SEZ6 is highly expressed in ASCL1- and NEUROD1-driven neuroendocrine subtypes and is currently under investigation as an ADC target, with ABBV-706 in early-phase trials. B7-H3 (CD276), overexpressed in non-neuroendocrine and NE-low subtypes, is being targeted by agents such as DS-7300a and the monoclonal antibody enoblituzumab. TROP2, though less commonly expressed in SCLC than in other epithelial tumors, remains of interest, with sacituzumab govitecan under evaluation. CEACAM5, expressed in a minority of cases, is being targeted by tusamitamab ravtansine in biomarker-selected populations. These efforts, supported by international thoracic oncology experts, reflect a growing shift toward precision medicine in SCLC.

### 2.10. Transformed Small-Cell Lung Cancer (t-SCLC)

Non-small-cell lung cancer (NSCLC) harboring epidermal growth factor receptor (EGFR) mutations may undergo histological transformation into small-cell lung cancer (SCLC) as a mechanism of acquired resistance to EGFR-tyrosine kinase inhibitors (EGFR-TKIs). Although transformed SCLC (t-SCLC) shares several histopathological features with de novo SCLC—including a high nuclear-to-cytoplasmic ratio, expression of neuroendocrine markers, and frequent inactivation of tumor suppressor genes RB1 and TP53—it is increasingly recognized as a distinct clinical entity. Patients diagnosed with t-SCLC generally have a poor prognosis. No defined subgroup appears to experience superior outcomes, with the exception of individuals in whom the interval between initial NSCLC diagnosis and subsequent SCLC transformation exceeds 12 months. Evidence regarding treatment efficacy in this subset is limited and derived primarily from retrospective analyses. Current data suggest that the addition of immune checkpoint inhibitors (ICIs) or EGFR-TKIs to platinum-based chemotherapy does not confer a significant survival advantage [23,24,25].

### 2.11. Future Perspectives in Biomarker Discovery for SCLC

Recent advances in omics technologies have begun to reshape our understanding of disease biology, particularly in the context of cancer. For example, while current biomarker strategies in small-cell lung cancer (SCLC) face significant limitations—including intratumoral heterogeneity, limited tissue availability, and dynamic expression profiles—emerging approaches such as single-cell RNA sequencing, proteomics, and spatial transcriptomics offer a promising path forward. These techniques are unveiling the intricate cellular architecture and micro-environmental interactions that underlie treatment resistance and immune evasion. Integrating these methodologies into biomarker discovery pipelines may enable more precise patient stratification and guide rational therapeutic combinations. Although still in their early stages, these technologies represent a transformative frontier in biomedical research. Their continued development and integration could profoundly impact our ability to decode cellular states and heterogeneity in complex biological systems, ultimately paving the way for more personalized and effective clinical interventions in the near future [26,27,28].

## 3. Conclusions

These developments support a shift toward precision oncology in SCLC, where biomarker-driven stratification could guide the selection of targeted therapies and improve outcomes in a historically aggressive and treatment-resistant malignancy. While the expanding therapeutic landscape offers new hope, it also raises the risk of fragmentation without a clear strategy for matching treatments to the right patients. Rather than a competitive race to dominate treatment lines, the true challenge lies in deepening our understanding of SCLC biology. A coordinated effort to define molecular subtypes and immune phenotypes could shift the paradigm from empirical treatment to precision oncology. Having multiple therapeutic options is undoubtedly a strength—but the goal must be to tailor therapy based on individual tumor characteristics, ensuring that each patient receives the most effective and personalized care possible.

## Figures and Tables

**Figure 1 curroncol-32-00536-f001:**
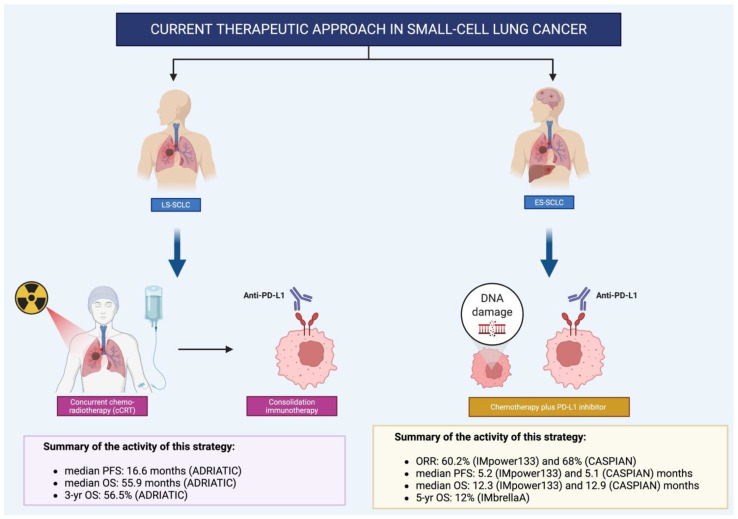
Current therapeutic approach in limited stage and extensive stage SCLC. Credit: Created with BioRender.com.

**Figure 2 curroncol-32-00536-f002:**
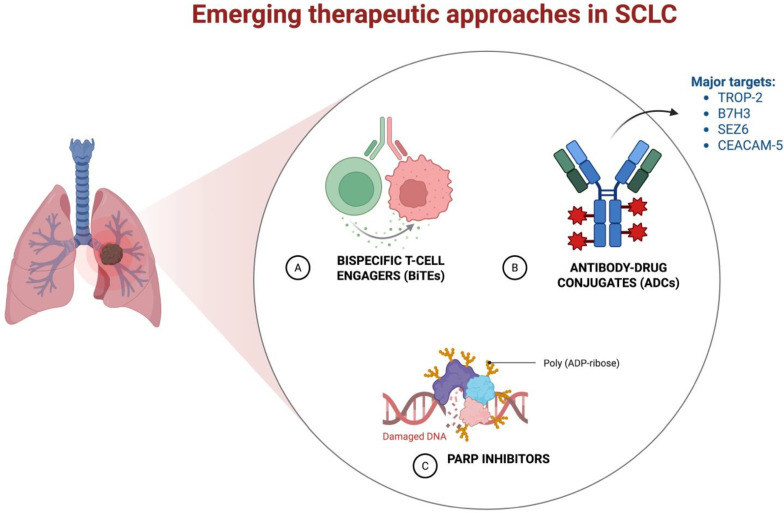
Emerging therapeutic agents in SCLC. Credit: Created with BioRender.com.

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
