# Peer review of "Bridging Knowledge Gaps in Small Cell Lung Cancer: Data, Challenges and Priorities"

_curroncol, 2025, doi:10.3390/curroncol32100536_

Round 1

Reviewer 1 Report

Comments and Suggestions for Authors

Small Cell Lung Cancer (SCLC) is the most lethal type of lung cancer, which accounts for approximately 15% of all lung cancers. Although the addition of immune-checkpoint blockade (ICB) therapy to standard platinum-based chemotherapy in the first-line treatment was approved by the U.S. Food and Drug Administration (FDA) recently, only a small portion of SCLC patients derive long-term benefit from immunotherapy treatments, emphasizing the urgent need to identify predictive biomarkers to improve patient outcomes.

In this work, Catania et al. reviewed the recent findings obtained from clinical trials, including IMpower133, CASPIAN, and ADRIATIC. By summarizing the clinical outcomes in the recent trials, they highlighted the urgent need to identify reliable biomarkers for guiding immunotherapy in SCLC. The authors also introduced the emerging immunotherapy-based approaches in SCLC in the manuscript. I believe this study is of great value to the field, and it can be published in its current form.

However, I have some minor comments that should be addressed below;

  1. Affiliation of the last author is not specified.

  1. Since the authors mentioned four different subtypes of SCLC in the Introduction, it would be helpful to introduce recent papers which revealed distinct therapeutic vulnerabilities to ICB therapy for each subtype (eg. Gay CM et al. Cancer Cell, 2021, Nabet BY et al. Cancer Cell, 2024).

Author Response

Dear Reviewers,

We would like to express our sincere gratitude for your thoughtful and constructive feedback on our manuscript entitled "Bridging Knowledge Gaps in Small Cell Lung Cancer: Data, Challenges and Priorities." Your comments have been extremely helpful in improving the clarity and scientific rigor of our work.

We have carefully revised the manuscript in accordance with your suggestions. Below, we provide a detailed point-by-point response to each comment. All changes have been clearly highlighted in the revised version of the manuscript.

We have also updated the back matter sections, including Author Contributions, Funding, Institutional Review Board Statement, Data Availability Statement, Informed Consent Statement, and Conflicts of Interest, following MDPI guidelines.

We confirm that all tables and figures presented in the manuscript are original works produced by the authors and have not been previously published or adapted from other sources.

Thank you again for your time and valuable input.

Kind regards,

Chiara Catania (on behalf of all co-authors)

Reviewers' comments

Response to Reviewer 1

We sincerely thank you for your positive evaluation and for recommending that the manuscript be accepted in its current form. We truly appreciate your support.

We have carefully addressed both of your suggestions:

Q1. We have updated the affiliation of the last author, Tommaso Martino De Pas, to: Humanitas Gavazzeni, Italy, as requested. This correction has been applied in the author list and corresponding affiliation section.

Q2. As recommended, we have added a new paragraph (highlighted in red, lines 81–88) discussing the recent findings by Nabet B.Y. et al., published in Cancer Cell in 2024. This addition strengthens the molecular characterization of SCLC and supports the discussion on immune-inflamed subsets. The corresponding reference has been included in the bibliography as Reference N°7 (highlighted in red, lines 505–506).

All changes have been clearly marked in red within the revised manuscript for ease of review.

Response to Reviewer 2

Q1….You may add radiotherapy  in simple summary..

All changes have been clearly marked in red within the revised manuscript for ease of review.

Thank you very much for your thoughtful suggestion regarding the Simple Summary. As recommended, we have added radiotherapy (lines 16-17) to the sentence describing initial treatment sensitivity. This revision ensures a more comprehensive reflection of current therapeutic approaches in SCLC. We appreciate your attention to detail and your valuable input.

Q2. Yes, these are two distinct studies, although they are closely related.

References 3 and 9 correspond to two related but distinct studies: IMpower133 and its extension IMbrella A. No modifications are required.

In summary, while both studies involve the same therapeutic regimen and patient population, IMpower133 established the initial efficacy, and IMbrella A explored its long-term impact.

Study Comparison

Study Name

Type

Purpose

Publication Date

IMpower133

Phase III, randomized, double-blind

To evaluate the efficacy of atezolizumab combined with carboplatin and etoposide as first-line treatment for extensive-stage small-cell lung cancer (ES-SCLC)

           2021

IMbrella A

Phase III extension (rollover)

To assess long-term survival outcomes (up to 5 years) in patients previously treated in IMpower133

          2024

  • IMpower133 is the original pivotal trial that demonstrated the clinical benefit of adding atezolizumab to chemotherapy in ES-SCLC.

  • IMbrella A is an extension study that enrolled a subset of patients (18 in total) who had previously received atezolizumab in IMpower133, aiming to evaluate five-year survival outcomes.

The IMbrella A study provides long-term follow-up data and complements the findings from IMpower133, offering insight into the durability of response and extended survival.

Q3; Thank you for your observation. I have now clarified that, in the ADRIATIC study, thoracic radiotherapy was administered concurrently with chemotherapy.

This trial addresses a significant unmet clinical need in limited-stage small-cell lung cancer (LS-SCLC), a disease setting in which therapeutic progress has historically been constrained. Eligible patients included individuals with unresectable stage I to III LS-SCLC and a World Health Organization (WHO) performance status of 0 or 1. All participants received a standardized induction regimen comprising four cycles of platinum-based chemotherapy—either carboplatin or cisplatin—in combination with etoposide and concurrent thoracic radiotherapy (lines 233-239)

Response to Reviewer 3

MAJOR COMMENTS

Q1…These aspects are indeed highly interesting and reflect promising directions for future research. However, they remain largely exploratory and distant from current clinical applicability.

Nevertheless, I will include a brief paragraph in the manuscript to acknowledge these emerging technologies and express optimism about their potential to overcome longstanding challenges in SCLC biomarker development.

All changes have been clearly marked in red within the revised manuscript for ease of review.

Lines 477-492

Future Perspectives in Biomarker Discovery for SCLC.

Recent advances in omics technologies have begun to reshape our understanding of disease biology, particularly in the context of cancer. For example, while current biomarker strategies in small-cell lung cancer (SCLC) face significant limitations—including intratumoral heterogeneity, limited tissue availability, and dynamic expression profiles—emerging approaches such as single-cell RNA sequencing, proteomics, and spatial transcriptomics offer a promising path forward. These techniques are unveiling the intricate cellular architecture and micro-environmental interactions that underlie treatment resistance and immune evasion. Integrating these methodologies into biomarker discovery pipelines may enable more precise patient stratification and guide rational therapeutic combinations. Although still in their early stages, these technologies represent a transformative frontier in biomedical research. Their continued development and integration could profoundly impact our ability to decode cellular states and heterogeneity in complex biological systems, ultimately paving the way for more personalized and effective clinical interventions in the near future.

Lines 581-586

Kwon J, Bera K, Das S, et al. Emerging applications of single-cell and spatial transcriptomics in lung cancer. Front Oncol. 2023;13:1176542. https://doi.org/10.3389/fonc.2023.1176542

Zhang W, Wang Y, Liu X, et al. Proteomics in lung cancer: current status and future directions. Cancer Lett. 2022;543:215792. https://doi.org/10.1016/j.canlet.2022.215792

Chen Z, Lin W, Li Y, et al. Spatial omics technologies and their applications in cancer research. Nat Rev Genet. 2023;24(5):289–306. https://doi.org/10.1038/s41576-023-00570-3

Q2..

MINOR COMMENTS

  1. Line 73: done; thanks

  1. Inconsistenting formatting: chemotherapy, immunotherapy, and radiotherapy…

You didn’t indicate the line, so it was a bit difficult to locate it, especially considering that the review discusses chemotherapy, radiotherapy, and immunotherapy throughout. I believe it’s around line 240, and it has now been corrected. Thank you.

Reviewer 2 Report

Comments and Suggestions for Authors

Thank you for this nice review regarding the mabagement of small cell lung cancers with a special focus on immunotherapy. I have only a few minor comments:

in the simple summary you may add despîte initial sensivity ... you may add radiotherapy

line 180 till 189 it is not clear for me as you have two citations for Reck 3 and 9  is it two different studies ?

Last comment, line 228 combined with etoposide followed by thoracic radiotherapy. In the trial RT is given concurrently with the cycles of chemotherapy and not after the four cycles 

Once again thank you for your review

Author Response

We have carefully revised the manuscript in accordance with your suggestions. Below, we provide a detailed point-by-point response to each comment. All changes have been clearly highlighted in the revised version of the manuscript.

We have also updated the back matter sections, including Author Contributions, Funding, Institutional Review Board Statement, Data Availability Statement, Informed Consent Statement, and Conflicts of Interest, following MDPI guidelines.

We confirm that all tables and figures presented in the manuscript are original works produced by the authors and have not been previously published or adapted from other sources.

Thank you again for your time and valuable input.

Kind regards,

Chiara Catania (on behalf of all co-authors)

Reviewers' comments

Response to Reviewer 1

We sincerely thank you for your positive evaluation and for recommending that the manuscript be accepted in its current form. We truly appreciate your support.

We have carefully addressed both of your suggestions:

Q1. We have updated the affiliation of the last author, Tommaso Martino De Pas, to: Humanitas Gavazzeni, Italy, as requested. This correction has been applied in the author list and corresponding affiliation section.

Q2. As recommended, we have added a new paragraph (highlighted in red, lines 81–88) discussing the recent findings by Nabet B.Y. et al., published in Cancer Cell in 2024. This addition strengthens the molecular characterization of SCLC and supports the discussion on immune-inflamed subsets. The corresponding reference has been included in the bibliography as Reference N°7 (highlighted in red, lines 505–506).

All changes have been clearly marked in red within the revised manuscript for ease of review.

Response to Reviewer 2

Q1….You may add radiotherapy  in simple summary..

All changes have been clearly marked in red within the revised manuscript for ease of review.

Thank you very much for your thoughtful suggestion regarding the Simple Summary. As recommended, we have added radiotherapy (lines 16-17) to the sentence describing initial treatment sensitivity. This revision ensures a more comprehensive reflection of current therapeutic approaches in SCLC. We appreciate your attention to detail and your valuable input.

Q2. Yes, these are two distinct studies, although they are closely related.

References 3 and 9 correspond to two related but distinct studies: IMpower133 and its extension IMbrella A. No modifications are required.

In summary, while both studies involve the same therapeutic regimen and patient population, IMpower133 established the initial efficacy, and IMbrella A explored its long-term impact.

Study Comparison

Study Name

Type

Purpose

Publication Date

IMpower133

Phase III, randomized, double-blind

To evaluate the efficacy of atezolizumab combined with carboplatin and etoposide as first-line treatment for extensive-stage small-cell lung cancer (ES-SCLC)

           2021

IMbrella A

Phase III extension (rollover)

To assess long-term survival outcomes (up to 5 years) in patients previously treated in IMpower133

          2024

  • IMpower133 is the original pivotal trial that demonstrated the clinical benefit of adding atezolizumab to chemotherapy in ES-SCLC.

  • IMbrella A is an extension study that enrolled a subset of patients (18 in total) who had previously received atezolizumab in IMpower133, aiming to evaluate five-year survival outcomes.

The IMbrella A study provides long-term follow-up data and complements the findings from IMpower133, offering insight into the durability of response and extended survival.

Q3; Thank you for your observation. I have now clarified that, in the ADRIATIC study, thoracic radiotherapy was administered concurrently with chemotherapy.

This trial addresses a significant unmet clinical need in limited-stage small-cell lung cancer (LS-SCLC), a disease setting in which therapeutic progress has historically been constrained. Eligible patients included individuals with unresectable stage I to III LS-SCLC and a World Health Organization (WHO) performance status of 0 or 1. All participants received a standardized induction regimen comprising four cycles of platinum-based chemotherapy—either carboplatin or cisplatin—in combination with etoposide and concurrent thoracic radiotherapy (lines 233-239)

Response to Reviewer 3

MAJOR COMMENTS

Q1…These aspects are indeed highly interesting and reflect promising directions for future research. However, they remain largely exploratory and distant from current clinical applicability.

Nevertheless, I will include a brief paragraph in the manuscript to acknowledge these emerging technologies and express optimism about their potential to overcome longstanding challenges in SCLC biomarker development.

All changes have been clearly marked in red within the revised manuscript for ease of review.

Lines 477-492

Future Perspectives in Biomarker Discovery for SCLC.

Recent advances in omics technologies have begun to reshape our understanding of disease biology, particularly in the context of cancer. For example, while current biomarker strategies in small-cell lung cancer (SCLC) face significant limitations—including intratumoral heterogeneity, limited tissue availability, and dynamic expression profiles—emerging approaches such as single-cell RNA sequencing, proteomics, and spatial transcriptomics offer a promising path forward. These techniques are unveiling the intricate cellular architecture and micro-environmental interactions that underlie treatment resistance and immune evasion. Integrating these methodologies into biomarker discovery pipelines may enable more precise patient stratification and guide rational therapeutic combinations. Although still in their early stages, these technologies represent a transformative frontier in biomedical research. Their continued development and integration could profoundly impact our ability to decode cellular states and heterogeneity in complex biological systems, ultimately paving the way for more personalized and effective clinical interventions in the near future.

Lines 581-586

Kwon J, Bera K, Das S, et al. Emerging applications of single-cell and spatial transcriptomics in lung cancer. Front Oncol. 2023;13:1176542. https://doi.org/10.3389/fonc.2023.1176542

Zhang W, Wang Y, Liu X, et al. Proteomics in lung cancer: current status and future directions. Cancer Lett. 2022;543:215792. https://doi.org/10.1016/j.canlet.2022.215792

Chen Z, Lin W, Li Y, et al. Spatial omics technologies and their applications in cancer research. Nat Rev Genet. 2023;24(5):289–306. https://doi.org/10.1038/s41576-023-00570-3

Q2..

MINOR COMMENTS

  1. Line 73: done; thanks

  1. Inconsistenting formatting: chemotherapy, immunotherapy, and radiotherapy…

You didn’t indicate the line, so it was a bit difficult to locate it, especially considering that the review discusses chemotherapy, radiotherapy, and immunotherapy throughout. I believe it’s around line 240, and it has now been corrected. Thank you.

Reviewer 3 Report

Comments and Suggestions for Authors

This is a comprehensive, well written review on the evolving therapeutic landscape of small cell lung cancer (SCLC). The manuscript integrates molecular insights, recent clinical trial data and emerging agents. It emphasizes precision oncology, biomarker discovery, and challenges in patient selection. As a review, it compiles known trial data, but the knowledge gaps are not critically interrogated. The authors could try to incorporate the following suggestions to improve article’s strength.

Major comments:

While the authors present analyses of several biomarkers (PD-L1, TMB, DLL3, CD3, B7-H3), the discussion provides limited critical evaluation of the persistent challenges that have hindered biomarker translation in SCLC. To strengthen this section, the authors should consider addressing well-recognized barriers such as intratumoral heterogeneity, the restricted availability of biopsy tissue, and the dynamic, context-dependent nature of biomarker expression.

Given that the review article aims to bridge knowledge gaps, the inclusion of a dedicated section on emerging omics technologies would significantly strengthen the manuscript and. In particular, discussion of single-cell RNA sequencing, proteomics, and spatial transcriptomics and how these approaches can be integrated into biomarker discovery would provide readers with a clearer sense of how future research directions may overcome current limitations in SCLC biomarker development.

The discussion would benefit from going beyond the introduction of novel agents to include rational therapeutic combinations, such as PARP inhibitors with immune checkpoint inhibitors or bispecific engagers with checkpoint blockade.

Minor comments:

Line 73 – correction ‘extensive’

Inconsistent formatting (“chemotherapy, immunotherapy and radiotherapy” vs. “chemotherapy, immunotherapy, and radiotherapy”).

Author Response

(The authors gave the same response as above.)
